# Lack of Association between YEASTONE Antifungal Susceptibility Tests and Clinical Outcomes of *Cryptococcus* Meningitis

**DOI:** 10.3390/jof9020232

**Published:** 2023-02-10

**Authors:** Ting-Shu Wu, Jung-Fu Lin, Chun-Wen Cheng, Po-Yen Huang, Jeng-How Yang

**Affiliations:** 1Division of Infectious Diseases, Department of Internal Medicine, Chang Gung Memorial Hospital at Linkou, College of Medicine, Chang Gung University, Taoyuan 33375, Taiwan; 2Infection Control Committee, Chang Gung Memorial Hospital at Linkou, Taoyuan 33375, Taiwan; 3Division of Infectious Diseases, Department of Internal Medicine, New Taipei Municipal TuCheng Hospital (Built and Operated by Chang Gung), New Taipei City 23652, Taiwan

**Keywords:** antifungal susceptibility, *Cryptococcus* meningitis, YEASTONE, amphotericin B, fluconazole

## Abstract

The relation between antifungal susceptibility and treatment outcomes is not well-characterized. There is paucity of surveillance data for cerebrospinal fluid (CSF) isolates of *cryptococcus* investigated with YEASTONE colorimetric broth microdilution susceptibility testing. A retrospective study of laboratory-confirmed cryptococcus meningitis (CM) patients was conducted. The antifungal susceptibility of CSF isolates was determined using YEASTONE colorimetric broth microdilution. Clinical parameters, CSF laboratory indices, and antifungal susceptibility results were analyzed to identify risk factors for mortality. High rates of resistance to fluconazole and flucytosine were observed in this cohort. Voriconazole had the lowest MIC (0.06 µg/mL) and lowest rate of resistance (3.8%). In a univariate analysis, hematological malignancy, concurrent cryptococcemia, high Sequential Organ Failure Assessment (SOFA) score, low Glasgow coma scale (GCS) score, low CSF glucose level, high CSF cryptococcal antigen titer, and high serum cryptococcal antigen burden were associated with mortality. In a multivariate analysis, meningitis with concurrent cryptococcemia, GCS score, and high CSF cryptococcus burden, were independent predictors of poor prognosis. Both early and late mortality rates were not significantly different between CM wild type and non-wild type species.

## 1. Introduction

*Cryptococcosis* is an infectious disease with worldwide distribution and a wide array of clinical presentations, including meningitis and disseminated disease [1]. Worldwide, nearly 220,000 new cases of cryptococcal meningitis (CM) occur each year, resulting in an estimated 181,000 deaths [2]. Pharmacological management of CM usually consists of primary therapy with amphotericin B, with or without flucytosine, followed by fluconazole maintenance therapy [3]. A regimen comprising amphotericin B or fluconazole is the preferred initial therapy for CM. There appears to be some correlation between minimal inhibitory concentration (MIC) and clinical resistance [4]. However, the guidelines of the Infectious Diseases Society of America do not suggest routine in vitro susceptibility testing of antifungal drugs in such cases [2]. Moreover, several reports have described the emergence of fluconazole-resistant *cryptococcus* and raised concerns regarding the widespread use of fluconazole in maintenance therapy for cryptococcal infection [5,6,7]. A reliable estimation of the antifungal susceptibility of CM isolates and an assessment of the correlation of the MIC of amphotericin B or fluconazole with the outcomes of CM are key objectives. Methods for the in vitro susceptibility testing of *C. neoformans* and *C. gattii* have been modified and standardized [8]. However, the value of MIC obtained by YEASTONE and its correlation with early and late outcomes of CM remain uncertain [9]. We utilized Thermo Fisher Scientific Sensititre YEASTONE colorimetric broth microdilution plates coupled with a Vizion Digital MIC Viewing System (a computer-assisted optical reading machine) to determine the in vitro susceptibility of cerebrospinal isolates of cryptococcus to amphotericin B, flucytosine, fluconazole, itraconazole, posaconazole, and voriconazole. This study aimed to assess the correlation between the antifungal susceptibility patterns of cerebrospinal *cryptococcus* isolates and to identify the risk factors for mortality in patients with CM.

## 2. Material and Methods

### 2.1. Study Population and Data Collection

We reviewed 53 patients with CM confirmed by CSF culture between 1 January 2010 and 31 December 2016. Of the 53 patients, 25 had concomitant cryptococcemia. The results of *cryptococcus* species identification, MICs of antifungal agents, and underlying comorbidities were analyzed. SOFA score was used to assess disease severity, and GCS score was used to evaluate the consciousness level. Laboratory results of CSF were collected. A cerebrospinal fluid cryptococcal antigen titer and serum antigen titer were used as indices of cryptococcus burden. Cryptococcus burden was defined as a logarithm of cryptococcus antigen to the base of 2. The treatment outcomes were 14-day mortality and overall in-hospital mortality.

### 2.2. Antifungal Susceptibility Testing

The MICs of antifungal agents were determined by SENSITITRE YEASTONE^®^, a colorimetric broth microdilution method for in vitro susceptibility tests. The YEASTONE microdilution plates were set up following the manufacturer’s instructions [9]. As no thresholds have been established for *cryptococcus* species, the epidemiological cutoff values (ECVs) used were based on the CLSI guidelines.

### 2.3. Cryptococcus Species Identification

*Cryptococcus* species were identified using matrix-assisted laser desorption ionization–time-of-flight mass spectrometry (MALDI-TOF MS) following the manufacturer’s recommendations [10,11,12].

### 2.4. Statistical Analysis

Statistical analysis was conducted using the SPSS software (IBM SPSS Statistics for Windows version 21.0; IBM Corp, NY, USA). Continuous variables were compared using the Mann–Whitney U-test for 2 groups. When the expected number of patients in any cell was less than 5, the categorical variables were compared using either the Chi-squared test or Fisher’s exact test. Correlations between continuous variables were assessed using the Pearson’s correlation coefficient. Risk factors associated with clinical outcomes were fitted in a logistic regression model. All statistical tests were 2-tailed, and *p*-values < 0.05 were considered indicative of statistical significance.

## 3. Results

### 3.1. Clinical Characteristics of the Study Population

A total of 53 patients with CM were enrolled in this study. The pathogenic species included 48 (90.6%) isolates of *C. neoformans*, 4 (7.5%) isolates of *C*. *gattii*, and 1 (1.9%) isolate of *C. curvatus*. The baseline demographic and clinical characteristics of the study population are summarized in Table 1. Of these, 12 (22.6%) patients were HIV-positive. Among the 53 meningitis patients, 25 (47.2%) had concurrent cryptococcemia. Three isolates (5.7%) of cryptococcus species were non-wild type isolates to amphotericin B (MIC > 0.5 µg/mL). Sixteen isolates (30.2%) were resistant to fluconazole (MIC > 8 µg/mL). The mortality rate during the 14-day induction therapy was 17%. The overall in-hospital mortality rate in this cohort was 50.9%.

### 3.2. MIC of Antifungal Agents

The MIC_50_, MIC_90_, range, epidemiologic cutoff values, and wild type versus non-wild type rates are presented in Table 2. The widest range of MIC was found in flucytosine (0.5–128 µg/mL). Voriconazole had the lowest MIC (0.06 µg/mL) and lowest rate of resistance (3.8%). A high prevalence of non-wild type resistance was observed for both fluconazole and flucytosine (30.2% and 34%, respectively).

### 3.3. Antifungal Susceptibility and Mortality Outcomes

Figure 1 depicts the correlation between MICs of amphotericin B and mortality outcomes. The 14-day mortality rates among amphotericin B MICs of 0.125 µg/mL, 0.25 µg/mL, 0.5 µg/mL, and 1 µg/mL were 0%, 25%, 14%, and 33%, respectively (Chi-squared test for trend *p* = 0.93). The overall in-hospital mortality rates associated with amphotericin B MICs of 0.125 µg/mL, 0.25 µg/mL, 0.5 µg/mL, and 1 µg/mL were 100%, 50%, 51%, and 33%, respectively (Chi-squared test for trend *p* = 0.37). The 14-day mortality rates and the overall in-hospital mortality rates among wild type and non-wild type were 16% vs. 33% (*p* = 0.47) and 52% vs. 33% (*p* = 0.7), respectively. There was no significant trend or statistical correlation between amphotericin B susceptibility and mortality outcomes. A similar result was observed for fluconazole (Figure 2). For fluconazole, the 14-day mortality rates and the overall in-hospital mortality rates among wild type vs non-wild type were 16% vs. 19% (*p* = 0.85) and 54% vs. 44% (*p* = 0.49), respectively.

### 3.4. Poor Prognostic Factors for 14-Day Mortality

On univariate analysis (Table 3), CSF glucose level <5 mg/dL (11.3% vs. 55.6%, *p* = 0.008), higher CSF cryptococcal antigen burden (10 vs. 12, *p* = 0.002), and higher serum cryptococcal antigen burden (9 vs. 12, *p* = 0.007) were found to contribute to 14-day mortality (Table 3). Disease severity, as assessed by SOFA score (1.5 vs. 3, *p* = 0.13) or GCS score (15 vs. 13, *p* = 0.43), and delayed amphotericin B induction (6.8% vs. 0%, *p* = 0.42) were not found to have contributed to the prognosis of initial 2-week therapy.

### 3.5. Poor Prognostic Factors for Overall in-Hospital Mortality

The risk factors associated with in-hospital mortality (Table 4) in the univariate analysis included hematology malignancy (4 vs. 0, *p* = 0.03), concurrent cryptococcemia (23.1% vs. 70.4%, *p* = 0.001), higher SOFA score (1 vs. 4, p = 0.001), lower GCS score (15 vs. 13, *p* = 0.001), lower CSF glucose level <5 mg/dL (7.7% vs. 29.6%, *p* = 0.04), higher CSF cryptococcal antigen burden (9 vs. 11, *p* = 0.003), and higher serum cryptococcal antigen burden (8.5 vs. 11, *p* = 0.023).

### 3.6. Multivariate Analysis for Poor Prognostic Factors

For multiple logistic regression, the candidate risk factors associated with *p* < 0.05 in univariate analyzes were selected. Along with SOFA score, GCS score, and cryptococcemia were set as covariates. On multivariate analysis, low CSF glucose level < 5 mg/dL (*p* = 0.034, odds ratio (OR) = 0.075, 95% confidence interval (CI) 0.007–0.85), high CSF cryptococcus burden (*p* = 0.023, OR = 2.588, 1.141–5.87), and high serum cryptococcus burden (*p* = 0.031, OR = 1.791, 95% CI 1.053–3.04) were identified as independent risk factors for 14-day mortality. For overall in-hospital mortality, meningitis with concurrent cryptococcemia (*p* = 0.013, OR = 0.034, 95% CI 0.002–0.48), GCS score (*p* = 0.028, OR = 0.262, 95% CI 0.079–0.86), and high CSF cryptococcus burden (*p* = 0.032, OR = 2.145, 95% CI 1.069–4.3) were identified as independent risk factors for a poor prognosis (Table 5).

## 4. Discussion

*Cryptococcus neoformans* and *cryptococcus gattii* are encapsulated, heterobasidiomycetous fungi first identified from an environmental source in 1894 [13]. These were initially considered as rare opportunistic pathogens in immunocompromised human populations. However, cases of advanced cryptococcosis have remarkably increased during the past two decades. Most patients with invasive CM had underlying conditions, including HIV, prolonged corticosteroid usage, organ transplantation, hematology malignancy, and diabetes [14]. However, an estimated 20% cryptococcosis patients without HIV infection have no apparent underlying disease or risk factors [15]. In our study, only 22.6% of patients with CM were HIV-positive. Furthermore, only 28% patients in our study had diabetic mellitus, and 18.9% had systemic lupus erythematous. There were still 14.2% of CM patients with no underlying conditions, which is consistent with previous studies [16]. The most common species causing CM in our cohort was *Cryptococcus neoformans*, accounting for 90.6% patients.

The methods for in vitro susceptibility testing of cryptococcus species have been modified and the ECVs are well-established [10,17]. The purpose of ECVs for antifungal agents is to enable the early detection of emerging resistance. A global antifungal surveillance study, conducted between 1997 and 2007, documented a progressive increase in resistance to fluconazole among *C. neoformans* isolates (resistant rates 7.3–11.1%) [18]. The increasing trend of fluconazole resistance is more noticeable in Asia. In our study, we noticed an ominously high percentage of non-wild type strains toward fluconazole and flucytosine. In our cohort, the resistance rate to fluconazole was 30.2%, and the resistant rate to flucytosine was 34%. Another study conducted by Yi-Chun Chen et al. also reported similar rates of non-susceptibility in Southern Taiwan [6]. The reported resistance rate of flucytosine among cryptococcus isolates from Africa and Cambodia was approximately 1–2%, but ranged up to 7% [19]. Our study reported a 34% resistance rate of flucytosine. We believe this is the first case-series from Taiwan to report the prevalence of flucytosine resistance. The practice of flucytosine monotherapy in the treatment of invasive cryptococcosis should be particularly discouraged due to a high resistance.

The reports of in vitro susceptibility were reported as wild type and non-wild type based on the ECV. No clinical threshold is currently available for any antifungal agent. The role of the susceptibility test result as a predictor for early or late clinical outcomes remains unclear. A previous study showed some correlation between fluconazole MICs and the poor prognosis of CM [4,11,20]. However, we did not observe such correlation in our study. We analyzed the 14-day mortality as early outcome and overall in-hospital mortality as a late outcome and assessed its correlation with the individual MIC range of antifungal agents. With escalating fluconazole MIC range, we found no significant trend of increasing mortality. The mortality rates between wild type and non-wild type did not show any significant difference, both for amphotericin B and fluconazole. Likewise, in the multivariate analysis, antifungal susceptibility was not found to be an independent predictor of 14-day or in-hospital mortality. However, we did find an extremely high rate of mortality (50% for 14-day and 100% for in-hospital) for pathogens with fluconazole MIC ≥64 µg/mL. When treating CM patients with initial MICs of fluconazole >64 µg/mL, treatment failure may possibly be directly related to drug resistance. 

The clinical manifestations of CM are nonspecific and difficult to distinguish from those of meningitis due to other causes. The most important prognostic factor for successful treatment of cryptococcosis is the ability to control a patient’s underlying disease. Several studies have examined the prognostic factors of CM; however, the correlation between clinical manifestations and the prognosis remains unclear [20,21,22]. In our study, a low CSF glucose level (<5 mg/dL), high CSF cryptococcal antigen titers, and high serum cryptococcus antigen burden were independent risk factors for poor early prognosis. Patients’ underlying conditions such as DM, SLE, or malignancy and treatment modalities such as early amphotericin B induction were not related to early treatment outcomes. For late outcomes, we identified meningitis concurrent with cryptococcemia, a low GCS score, and high CSF cryptococcus burden as independent risk factors for poor prognosis.

## 5. Conclusions

In summary, CM may also occur in non-high-risk groups, such as patients without HIV infection or predisposing underlying conditions. The overall mortality rate in this cohort was high. Risk factors associated with mortality included concomitant cryptococcemia, low CSF glucose level, high CSF and serum cryptococcus antigen burden. Both early and late mortality rates were not significantly different between wild type and non-wild type species.

## Figures and Tables

**Figure 1 jof-09-00232-f001:**
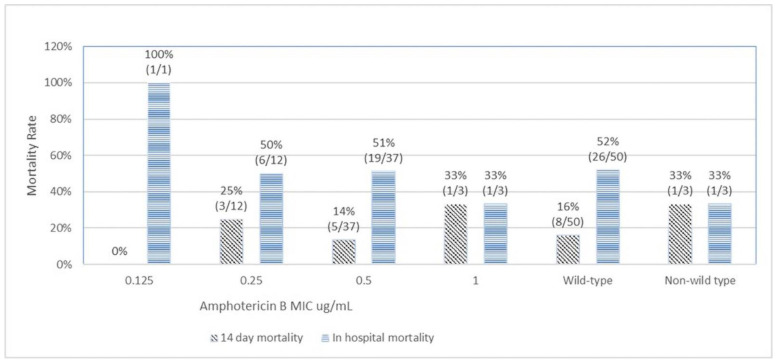
Correlation of amphotericin B minimal inhibitory concentration (MIC) with 14-day and in-hospital mortality.

**Figure 2 jof-09-00232-f002:**
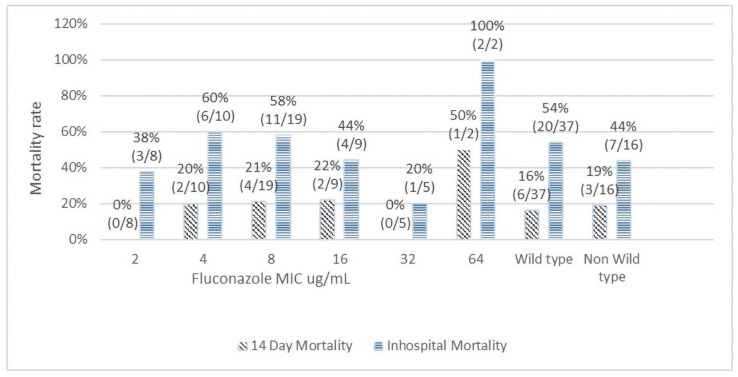
Correlation of fluconazole minimal inhibitory concentration with 14-day and in-hospital mortality.

**Table 1 jof-09-00232-t001:** Clinical characteristics, demographic data, and outcomes of patients with cryptococcal meningitis.

Variables	Value (IQR)
Age, years	54.3 (36.5–65.7)
Sex, male	17 (32.1%)
**Condition**	
HIV	12 (22.6%)
DM	11 (20.8%)
SLE	10 (18.9%)
Solid tumor	3 (5.7%)
Hematological disease	4 (7.5%)
ESRD	2 (3.8%)
Liver cirrhosis	4 (7.5%)
Cryptococcemia	25 (47.2%)
Pulmonary cryptococcus	7 (13.2%)
CT/MRI abnormality	39 (73.6%)
***Cryptococcus* species**	
*Cryptococcus curvatus*	1 (1.9%)
*Cryptococcus gattii*	4 (7.5%)
*Cryptococcus neoformans var grubii*	48 (90.6%)
**Disease severity**	
SOFA score	2 (0–4)
Glasgow coma scale score	15 (11.5–15.0)
Increased intracranial pressure signs	41 (77.4%)
**Initial laboratory data**	
CSF opening pressure (cmH_2_O)	29 (22–38.5)
CSF WBC count (/µL)	15 (3–81)
CSF glucose (mg/dL)	38 (9.5–54.5)
CSF protein (mg/dL)	91 (60–171)
CSF cryptococcus burden	10 (8–12)
Serum cryptococcus burden	10 (8–12)
Non-wild type isolate to amphotericin B (MIC > 0.5 µg/mL)	3 (5.7%)
Non-wild type isolate to fluconazole (MIC > 8 µg/mL)	16 (30.2%)
**Clinical outcome**	
14-day mortality	9 (17.0%)
30-day mortality	14 (26.4%)
In-hospital mortality	27 (50.9%)

Categorical variables are presented as frequency (%) and continuous variables are expressed as median (IQR; interquartile range). SOFA = Sequential Organ Failure Assessment; a scoring system for assessment of sepsis (from 0 to 24). Glasgow coma scale, a scoring system to evaluate the consciousness level including eye opening, verbal response, and motor response (from 3 to 15). DM = diabetic mellitus. SLE = systemic lupus erythematosus. CT = computerized tomography. MRI = magnetic resonance imaging. ESRD = end-stage renal disease. SOFA = Sequential Organ Failure Assessment. CSF = cerebrospinal fluid.

**Table 2 jof-09-00232-t002:** Distribution of the minimal inhibitory concentrations of antifungal agents among 53 cryptococcal meningitis CSF isolates.

	MIC (µg/mL)						
Antifungal Agents	Range	MIC_50_	MIC_90_	Modes	ECVs	Wild Type (%)	Non-Wild Type (%)
Amphotericin B	0.12–1.0	0.5	0.5	0.5	0.5	50 (94.3)	3 (5.7)
Fluconazole	2–64	8	32	8	8	37 (69.8)	16 (30.2)
Flucytosine	0.5–128	8	16	16	8	35 (66)	18 (34)
Itraconazole	0.03–0.5	0.12	0.25	0.12	0.25	51 (96.2)	2 (3.8)
Posaconazole	0.03–1.0	0.12	0.5	0.25	0.25	47 (88.7)	6 (11.3)
Voriconazole	0.015–0.5	0.06	0.25	0.06	0.25	51 (96.2)	2 (3.8)

ECVs: epidemiological cutoff values. MIC: minimal inhibitory concentration. MIC_50_: minimal inhibitory concentration for inhibition of 50% isolates. MIC_90_: minimal inhibitory concentration for inhibition of 90% isolates.

**Table 3 jof-09-00232-t003:** Factors associated with 14-day mortality of patients with cryptococcal meningitis.

	Survivor (N = 44)	Non-Survivor (N = 9)	Univariate *p*-Value
Age, years	55.3 (37.2–65.9)	38.1 (28.4–67.4)	0.23
Sex, male	30 (68.2%)	6 (66.7%)	0.92
**Condition**			
HIV infection	8 (18.2%)	4 (44.4%)	0.18
Diabetes mellitus	10 (22.7%)	1 (11.1%)	0.66
SLE	7 (15.9%)	3 (33.3%)	0.35
Solid tumor	2 (4.5%)	1 (11.1%)	0.43
Hematology malignancy	4 (9.1%)	0 (0%)	0.46
ESRD	2 (4.5%)	0 (0%)	0.68
Liver cirrhosis	3 (6.8%)	1 (11.1%)	0.53
Cryptococcemia	19 (43.2%)	6 (66.7%)	0.18
Pulmonary cryptococcus	5 (11.4%)	2 22.2%)	0.33
CT/MRI abnormality	32 (72.7%)	7 (77.8%)	0.55
**Disease severity**			
SOFA score	1.5 (0–4)	3 (0.5–8)	0.13
Glasgow coma scale score	15 (12.25–15)	13 (8.5–15)	0.43
Signs of increased intracranial pressure	35 (79.5%)	6 (66.7%)	0.41
**Initial laboratory data**			
CSF opening pressure >30 cmH_2_O	27 (61.3%)	4 (44.4%)	0.46
CSF WBC count (/µL)	30.5 (2.25–85.5)	10 (3–68)	0.99
CSF glucose <5 mg/dL	5 (11.3%)	5 (55.6%)	0.008 *
CSF protein >500 mg/dL	4 (9.1%)	1 (11.1%)	0.62
CSF cryptococcus burden	10 (8–11)	12 (11.5–14)	0.002 *
Serum cryptococcus burden	9 (8–11)	12 (11.5–13)	0.007 *
Non-wild type isolate to amphotericin B (MIC > 0.5 µg/mL)	2 (4.5%)	1 (11.1%)	0.43
Non-wild type isolate to fluconazole (MIC > 8 µg/mL)	13 (29.5%)	3 (33.3%)	0.55
**Treatment modality**			
Delayed Amphotericin B use ^a^	3 (6.8%)	0 (0%)	0.42

Categorical variables are presented as frequency (%); continuous variables are presented as median (interquartile range). SLE = Systemic Lupus Erythematosus. ESRD = End Stage Renal Disease. SOFA = Sequential Organ Failure Assessment. CSF = Cerebrospinal fluid. * *p* < 0.05. ^a^ Delayed amphotericin B use was defined as antifungal agent use beyond 3 days of fungal culture.

**Table 4 jof-09-00232-t004:** Factors associated with in-hospital mortality.

	Survivor (N = 26)	Non-Survivor (N = 27)	Univariate *p*-Value
Age, years	56.5 (36.5–64.7)	46.7 (34.1–73.4)	0.78
Sex, male	21 (80.8%)	15 (55.6%)	0.05
**Condition**			
HIV	4 (15.4%)	8 (29.6%)	0.22
Diabetic mellitus	2 (7.7%)	9 (33.3%)	0.22
SLE	3 (11.5%)	7 (25.9%)	0.18
Solid tumor	1 (3.8%)	2 (7.4%)	0.58
Hematology malignancy	4 (15.4%)	0 (0%)	0.03
ESRD	0 (0%)	2 (7.4%)	0.16
Liver cirrhosis	1 (3.8%)	3 (11.1%)	0.32
Cryptococcemia	6 (23.1%)	19 (70.4%)	0.001 *
Pulmonary cryptococcus	3 (11.5%)	4 (14.8%)	0.73
CT/MRI abnormality	18 (69.2%)	21 (77.8%)	0.49
**Disease severity**			
SOFA score	1 (0–2)	4 (1–7)	0.001 *
Glasgow coma scale score	15 (14.75–15)	13 (8.0–15)	0.001 *
Increased intracranial pressure	22 (84.6%)	19 (70.4%)	0.22
**Initial laboratory data**			
CSF opening pressure >30 cmH2O	15 (57.7%)	16 (59.3%)	0.91
CSF WBC count (/µL)	36.5 (2.75–93.75)	13 (3–46)	0.41
CSF glucose <5 mg/dL	2 (7.7%)	8 (29.6%)	0.041 *
CSF protein >500 mg/dL	1 (3.8%)	4 (14.8%)	0.17
CSF cryptococcus burden	9 (6.75–11)	11 (10–12)	0.003 *
Serum cryptococcus burden	8.5 (6–10.25)	11 (8v12)	0.023 *
Non-wild type isolate to amphotericin B (MIC > 0.5 µg/mL)	2 (7.7%)	1 (3.8%)	0.53
Non-wild type isolate to fluconazole (MIC > 8 µg/mL)	9 (34.6%)	7 (25.9%)	0.52

Categorical variables are presented as frequency (%); continuous variables are presented as median (interquartile range). SLE = systemic lupus erythematosus. ESRD = end-stage renal disease. SOFA = Sequential Organ Failure Assessment. CSF = cerebrospinal fluid. * *p* < 0.05.

**Table 5 jof-09-00232-t005:** Multivariate analysis of factors associated with unfavorable outcomes.

Variables	14-Day Mortality			In Hospital Mortality		
	OR	95% CI	*p*-Value	OR	95% CI	*p*-Value
**Clinical feature**						
Concurrent cryptococcemia				0.034	0.002–0.487	0.013 *
SOFA score				1.17	0.69–1.99	0.57
Glasgow coma scale				0.262	0.079–0.863	0.028 *
**Laboratory finding**						
CSF glucose <5 mg/dL	0.075	0.007–0.857	0.037 *	1.016	0.97–1.064	0.49
CSF cryptococcus burden	2.588	1.141–5.872	0.023 *	2.145	1.069–4.303	0.032 *
Serum cryptococcus burden	1.791	1.053–3.047	0.031^*^	0.915	0.6–1.395	0.678^*^

OR = odds ratio; CI = confidence interval; SOFA = Sequential Organ Failure Assessment; CSF = cerebrospinal fluid. * *p* < 0.05.

## Data Availability

Data is unavailable due to privacy or ethical restrictions.

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
