# Peer review of "Lack of Association between YEASTONE Antifungal Susceptibility Tests and Clinical Outcomes of Cryptococcus Meningitis"

_jof, 2023, doi:10.3390/jof9020232_

Round 1

Reviewer 1 Report (New Reviewer)

Dear Authors,

thank you for the opportunity to read this manuscript.

I have a few comments about the manuscript.

The aspect that (in my opinion) is missing here is

no connection between the aspect of the form of the drug used in the therapy and that in the case of the YEASTONE set. It should be added that gender also affects the metabolism of antifungal substances, which is the case for women and azole drugs. The missing element is that in the case of Amphotericin B, we are talking about a high-molecular compound that, depending on the form, has a different degree of penetration into the compartment, which is the cerebrospinal fluid. The same is true for fluconazole. Many publications lack this element. Comparing in the literature, the MIC values obtained, for example, for AmB, but depending on whether we have a pure compound, a liposomal form, a nanoform will be different.

Please refer to the current Cryptococcus taxonomy

https://www.mdpi.com/2309-608X/7/4/260

More current literature should be used.

Author Response

Dear reviewer, we thank for your comments and suggestions which is of great help for this work. We have modified our article with your suggestions. Please refer to revised version.

Our work here is trying to establish the connection between minimal inhibitory concentrations of antifungal agents and clinical outcomes. It is a traditional recognition that the higher mortality rate is associated with increased MIC levels of pathogens. However, predicting the clinical outcome of cryptococcal meningitis based solely on antifungal susceptibility is difficult because the prognosis of the disease itself depends on numerous factors (degree of immune suppression, severity of illness, and adherence to therapy), which could not be accounted for because of the retrospective nature. The heterogeneity of the patient population and yes, the differences in amphotericin B compound may more considerably affect the primary outcomes. We did not analyze the compounds of amphotericin and its effect for mortality because this is a short report focusing on YEASTONE susceptibility test. However, for those who treated with fluconazole (the compound is unified), in our data still showed no correlation between MIC and clinical outcomes.  With escalating fluconazole MIC range, we found no significant trend of increasing mortality. The mortality rates between wild type and non-wild type did not show any significant difference, both for amphotericin B and fluconazole. Likewise, on multivariate analysis, antifungal susceptibility was not found to be an independent predictor of 14-day or in-hospital mortality. We acknowledge different compounds of amphotericin may exert different penetration. But the YEASTONE MIC tests did not effect the overall outcome of mortality. Similar results were also noted by our study and has been published. (Reference : J.-H. Yang, P.-Y. Huang, C.-W. Cheng, S.-S. Shie, Z.-F. Lin, L.-Y. Yang, C.-H. Lee, T.-S. Wu, Antifungal susceptibility testing with YeastONETM is not predictive of clinical outcomes of Cryptococcus neoformans var. grubii fungemia, Med Mycol. 59 (2021) 1114–1121. https://doi.org/10.1093/mmy/myab046.)

Reviewer 2 Report (New Reviewer)

The paper reports the lack of correlation between non-wild type fluconazole and amphotericin B C. neoformans and mortality at 14 days and in-hospital stay.  There are several concerns about this analysis. The cut-off for non-wild type was taken from another susceptibility test with no reference cited to show comparability between YEASTONE the standard micro dilution assay.  The authors report Yeastone results for drugs not used the patients reported here :  flucytosine, posaconazole, itraconazole and voriconazole.  That data should be justified or omitted. The unusually high mortality (27 of 53 patients, or 51%) during hospitalization deserves comment in order to show why this case series is representative of other populations. The discussion on lines 225-230 concludes that patients with a fluconazole MIC by Yeastone should be given a more susceptible azale, such as voriconazole. This statement is based on two patients given fluconazole and none given voriconazole; this should be deleted. Cryptococcus curvets (Table 1) is a nonpathogen that has been moved out of the genus Cryptococcus.

There are multiple papers on risk factors for poor outcome in cryptococcal meningitis, the factors varying with the treatment given and the specific population. It is not clear what this paper contributes to the literature. What  did they learn that is new?

Author Response

Reply: Dear reviewer, we thank for your comments and suggestions which is of great help for this work. We have modified our article with your suggestions. Please refer to revised version.

The standard broth microdilution method according to US National Committee for Clinical Laboratory Standards document M27-A is usually recommended to determine the in vitro susceptibility to antifungal agents. However, K.G Davey compared the Sensitire Yeast One method with a standard broth microdilution test and found that the overall agreement between the results of the 2 methods were 99% for amphotericin B, 96.5% for flucytosine, and 91.5% for fluconazole[Reference K.G. Davey, A. Szekely, E.M. Johnson, D.W. Warnock, Comparison of a new commercial colorimetric microdilution method with a standard method for in-vitro susceptibility testing of Candida spp. and Cryptococcus neoformans., J Antimicrob Chemother. 42 (1998) 439–444. https://doi.org/10.1093/jac/42.4.439.].

The Sensitire Yeast One results showed high agreement with CLSI methods. Besides, a French cryptococcosis study group analyzing various antifungal susceptibility testing methods concluded that the antifungal susceptibility test results obtained by either the CLSI methods, E test, or broth microdilution in YNB medium did not predict early clinical outcomes in patients with cryptococcosis[Reference E. Dannaoui, M. Abdul, M. Arpin, A. Michel-Nguyen, M. Piens, A. Favel, O. Lortholary, F. Dromer, Results obtained with various antifungal susceptibility testing methods do not predict early clinical outcome in patients with cryptococcosis, Antimicrob Agents Chemother. 50 (2006) 2464–2470. https://doi.org/10.1128/AAC.01520-05. ].

Owing to this is a short brief report, we did not cite the reference in our discussion. But research on YEASTONE MIC is planty.

As for what is new about risk factors for poor outcome, in our study, low CSF glucose level (<5 mg/dL), high CSF cryptococcal antigen titers, and high serum cryptococcus antigen burden were independent risk factors for poor early prognosis. The patients’ underlying conditions such as DM, SLE, or malignancy and treatment modalities such as early amphotericin B induction were not related to early treatment outcomes. For late outcome, we identified meningitis concurrent with cryptococcemia, low GCS score, and high CSF cryptococcus burden as independent risk factors for poor prognosis.

Reviewer 3 Report (New Reviewer)

Author Yang et al. describe the "Lack of Association between YEASTONE Antifungal Susceptibility of Cryptococcus Meningitis Cerebrospinal Fluid Isolates and Risk Factors for Poor Prognosis".

Following are the comments;

1: The title is confusing 

2: The introduction is not convincing, it needs to elaborate with the clear objective and advantage of the proposed method.

3: Identification of Cryptococcus species should be made by sequencing the 16S rRNA gene.

4: What are the value of MBC and the ratio of MBC/MIC

5: The resolution of Figure 1 and 2 need to improve.

Author Response

Reviewer 3

Comments and Suggestions for Authors

Author Yang et al. describe the "Lack of Association between YEASTONE Antifungal Susceptibility of Cryptococcus Meningitis Cerebrospinal Fluid Isolates and Risk Factors for Poor Prognosis".

Following are the comments;

1: The title is confusing 

Reply: The title is revised and should be comprehensive now.

2: The introduction is not convincing, it needs to elaborate with the clear objective and advantage of the proposed method.

Reply: We thank for your comments and suggestions which is of great help for this work. We have modified our article with your suggestions. Please refer to revised version.

3: Identification of Cryptococcus species should be made by sequencing the 16S rRNA gene.

Reply: This study was conducted during Jan 1st, 2010 through Dec 31th, 2016 in single hospital. Duration the period, our hospital applied MALDI–TOF MS to identified Cryptococcus. The new technology of rep-PCR DNA fingerprinting was applied recently.

4: What are the value of MBC and the ratio of MBC/MIC

Rely: According to the ratio MBC/MIC, we appreciated antibacterial activity for anit-bacterial agents. However, no clinical threshold is currently available for any antifungal agents. Therefore, the reports of in vitro susceptibility for antifungal agents were reported as wild type and non-wild type based on the epidemiologic cut off. Since the true clinical cutoff is unclear, we do not certain the value of MBC and MBC/MIC ratio in clinical settings.

5: The resolution of Figure 1 and 2 need to improve.

Reply: the resolution is enhanced now.

Reviewer 4 Report (New Reviewer)

Jeng-How Yang et al describe the relation between antifungal susceptibility determined by the YEASTONE colorimetric broth microdilution susceptibility test and treatment outcomes of cryptococcosis to identify risk factors for mortality. The work is very interesting and the topic is of fundamental importance in diagnosing and choosing the best therapy. However, there are minor inaccuracies that need to be changed:

·         No data were described on the immunity status of the examined patients. Considering that the pathogen in question is an opportunist, some details should be added if possible.

·         Some references are ancient (number 1 is from 1983, and number 21 is from 1979). Are the authors sure there aren't more recent papers describing cryptococcosis or treatment with amphotericin and flucytosine?

·         Abstract: The acronym CM should be written in full the first time it appears in the text.

·         Page 2, line67: “Sensitire” should be replaced with “Sensititre”

·         Page 2, line 88: In C.gattii add the space

·         Page 3: line 89: In C.curvatus add the space

·         Table 1: Add the meaning of the DM and CT/MRI acronyms in the table legend.

·         Page 4, line 109: Which strains do the authors refer to with “wild type” and “non-wild type”?

·         Page 4, line 111: Voriconazole showed the lowest MIC (0.06 μg/mL): Do the authors mean the MIC50?

·         Page 4, lines 112-113: How the resistance assessment is performed should be explained.

·         Figures 1 and 2 could be merged.

·         The tables should be shown on the same page for ease of reading.

·         cryptococcus neoformans or C. neoformans should always be written in italics and with a capital letter C throughout the manuscript

Page 8, lines 217-218: “A previous study showed some correlation between fluconazole MICs and poor prognosis of CM” - Three references are cited, 4, 11, and 20. Which of the three is the study referring to?

Author Response

Jeng-How Yang et al describe the relation between antifungal susceptibility determined by the YEASTONE colorimetric broth microdilution susceptibility test and treatment outcomes of cryptococcosis to identify risk factors for mortality. The work is very interesting and the topic is of fundamental importance in diagnosing and choosing the best therapy. However, there are minor inaccuracies that need to be changed:

  • No data were described on the immunity status of the examined patients. Considering that the pathogen in question is an opportunist, some details should be added if possible.

Reply: We described the clinical conditions such as HIV status, and underling disease with diabetic mellitus or malignancy in table 1, which should indicate the immune status of patients. And in table 3 we also analyzed the above factors with mortality outcome.

  • Some references are ancient (number 1 is from 1983, and number 21 is from 1979). Are the authors sure there aren't more recent papers describing cryptococcosis or treatment with amphotericin and flucytosine?

Reply : yes, the standard treatment for cyrptococcosis had no breakthrough advances and the clinical practice guideline from infectious disease society of America has not been updated since 2010.

  • Abstract: The acronym CM should be written in full the first time it appears in the text.

Reply: Thanks for the reminding. The acronym is written now.

  • Page 2, line67: “Sensitire” should be replaced with “Sensititre”

Reply: The typing error is corrected.

  • Page 2, line 88: In C.gattiiadd the space

Reply: space added

  • Page 3: line 89: In C.curvatusadd the space

Reply: space added

  • Table 1: Add the meaning of the DM and CT/MRI acronyms in the table legend.

Reply: The acronym added

  • Page 4, line 109: Which strains do the authors refer to with “wild type” and “non-wild type”?

Reply: The wild type means susceptible to antifungal agents and then non-wild type means non-susceptible to antifungal agents.

  • Page 4, line 111: Voriconazole showed the lowest MIC (0.06 μg/mL): Do the authors mean the MIC50?

Reply: Yes, the MIC means MIC50.

  • Page 4, lines 112-113: How the resistance assessment is performed should be explained.

Reply: The resistance assessment is based on the CLSI guidelines and was explained in Material/method antifungal susceptibility test.

  • Figures 1 and 2 could be merged.

Reply: Amphotericin B and fluconazole remain the standard for cryptococcus treatments. We demonstrated the MIC of Amphotericin B and fluconazole and outcome for mortality in figure 1 and 2. These two figures can be merged but we think it would be more easier to read if we separate. 

  • The tables should be shown on the same page for ease of reading.

Reply: The position of tables is adjusted.

  • cryptococcus neoformans or C. neoformans should always be written in italics and with a capital letter C throughout the manuscript

Reply: The written is adjusted now

Page 8, lines 217-218: “A previous study showed some correlation between fluconazole MICs and poor prognosis of CM” - Three references are cited, 4, 11, and 20. Which of the three is the study referring to?

Reply: The 3 references study all concluded poor outcome with elevated MIC.

Round 2

Reviewer 2 Report (New Reviewer)

Except for deleting one erroneous sentence, none of the problems has been addressed. The top of FIG.2  seems to have acquired a new, red line across the graph.

Reviewer 3 Report (New Reviewer)

Accept

This manuscript is a resubmission of an earlier submission. The following is a list of the peer review reports and author responses from that submission.

Round 1

Reviewer 1 Report

1. In figure 1, why the mortality rate of in-hospital mortality of wild-type isolates demonstrated higher than non-wild type isolates? What are the factors? May you add these comments in the discussion section.

2. Please give more detail the SOFA score and GCS score in the result section on page 5 line 170. Are the lower- or higher score represent the poor prognosis?

3. Page 8, line 214, please use the italic letter for C. neoformans.

Author Response

Comments and Suggestions for Authors

  1. In figure 1, why the mortality rate of in-hospital mortality of wild-type isolates demonstrated higher than non-wild type isolates? What are the factors? May you add these comments in the discussion section.

Reply: Dear reviewer, first we want to thank you for your kind review and comment. With your suggestions, the revised version of our article is better than the original version. Here is my point to point reply to your comments.

The in-hospital mortality rate is numerically higher among wild type isolates than non-wild type isolates, however, the difference did not reach statistical significance level (p=0.62). This is one of our conclusions that the antifungal susceptibility test of cryptococcus is not predictive factor for mortality outcomes.

  1. Please give more detail the SOFA score and GCS score in the result section on page 5 line 170. Are the lower- or higher score represent the poor prognosis?

Reply: We have added detail description of the score system in article.

The Sequential Organ Failure Assessment (SOFA) score is a scoring system that assesses the performance of several organ systems in the body (neurologic, blood, liver, kidney, and blood pressure/hemodynamics) and assigns a score based on the data obtained in each category. The score is calculated based on Glasgow coma scale, mean arterial pressure or administration of vasopressors, PaO2/FiO2, platelet counts, bilirubin levels, and creatinine.

The higher the SOFA score, the higher the likely mortality.

The Glasgow Coma Score (GCS) is a commonly used index for evaluating the level of consciousness and overall status of the central nervous system. An initial score of less than 5 is associated with an 80% chance of being in a lasting vegetative state or death. 

The criteria for SOFA score are listed as below.

FORMULA

Addition of the selected points:

Variable

Points

PaO2/FiO2*, mmHg

≥400

0

300-399

+1

200-299

+2

≤199 and NOT mechanically ventilated

+2

100-199 and mechanically ventilated

+3

<100 and mechanically ventilated

+4

Platelets, ×103/µL

≥150

0

100-149

+1

50-99

+2

20-49

+3

<20

+4

Glasgow Coma Scale

15

0

13–14

+1

10–12

+2

6–9

+3

<6

+4

Bilirubin, mg/dL (μmol/L)

<1.2 (<20)

0

1.2–1.9 (20-32)

+1

2.0–5.9 (33-101)

+2

6.0–11.9 (102-204)

+3

≥12.0 (>204)

+4

Mean arterial pressure OR administration of vasoactive agents required (listed doses are in units of mcg/kg/min)

No hypotension

0

MAP <70 mmHg

+1

DOPamine ≤5 or DOBUTamine (any dose)

+2

DOPamine >5, EPINEPHrine ≤0.1, or norEPINEPHrine ≤0.1

+3

DOPamine >15, EPINEPHrine >0.1, or norEPINEPHrine >0.1

+4

Creatinine, mg/dL (μmol/L) (or urine output)

<1.2 (<110)

0

1.2–1.9 (110-170)

+1

2.0–3.4 (171-299)

+2

3.5–4.9 (300-440) or UOP <500 mL/day)

+3

≥5.0 (>440) or UOP <200 mL/day

+4

*Estimating FiOâ‚‚ from oxygen flow/delivery rates:

Type of Oâ‚‚ delivery

Flow rates, L/min

FiOâ‚‚

Nasal cannula

1-6

~4% FiOâ‚‚ added above room air* per 1 L/min

  • Room air = 21% 
  • 1 L/min = 25%
  • 2 L/min = 29%
  • 3 L/min = 33%
  • 4 L/min = 37%
  • 5 L/min = 41%
  • 6 L/min = 45%

Simple face mask

~6-12

35-60%*

Non-rebreather mask

10-15

~70-90%

High-flow nasal cannula

Up to 60

30-100%

  1. Page 8, line 214, please use the italic letter for C. neoformans.

Reply: Thank you for the reminding. The name of species is in italic letter now.

Reviewer 2 Report

In the paper by Yang and colleagues, the authors relate Cryptococcus meningitis and some factors associated with it, such as antifungal susceptibility, to the prognosis of the disease.

Work-related observations

Authors should correct and standardize the way they write the name of fungal species and genera. They should consider that the first letter of the genus should be capitalized and that they should correctly use italics to describe genera and species of fungi.

The results are presented with varying number of decimal places. Standardizing the number of decimal places would improve the quality of the work.

In Table 1 several acronyms were used but their meaning was not referred to in the table legend. This information should be included.

Author Response

In the paper by Yang and colleagues, the authors relate Cryptococcus meningitis and some factors associated with it, such as antifungal susceptibility, to the prognosis of the disease.

Work-related observations

  1. Authors should correct and standardize the way they write the name of fungal species and genera. They should consider that the first letter of the genus should be capitalized and that they should correctly use italics to describe genera and species of fungi.

Rely: Dear reviewer, first we want to thank you for your kind review and comment. With your suggestions, the revised version of our article is better than the original version. The name of fungal species is corrected and standardized now. Please refer to revised version of the article.

The results are presented with varying number of decimal places. Standardizing the number of decimal places would improve the quality of the work.

Reply: The number of decimals is standardized in revised version. Thank you for the suggestions.

In Table 1 several acronyms were used but their meaning was not referred to in the table legend. This information should be included.

Reply: table legend is added for acronyms. Thank you.

Reviewer 3 Report

The manuscript attempted to investigate the correlation of antifungal susceptibility with prognosis of patients with cryptococcal meningitis . The authors reported concomitant cryptococcemia, low CSF glucose level, high CSF and serum cryptococcus antigen burden were risk factors for mortality. CM patients with fluconazole MIC ≥64 μg/mL showed extremely high mortality. However, the manuscript is not suitable to publication in current form and need to be substantially revised, because there are several limitations of this study, as shown below:

Major points

1. The most major limitation of this study is that the sample size of isolates is too small to effectively reflect the relationship between antifungal susceptibility of isolates and prognosis.   

2. The overall in-hospital mortality rates associated with amphotericin B MICs of 0.125 μg/mL, 0.25 μg/mL, 0.5 28 μg/mL, and 1 μg/mL were 100%, 50%, 51%, and 33%, respectively. The overall in-hospital mortality rates associated with fluconazole MICs of 2 μg/mL, 4 μg/mL, 8 μg/mL, 16 μg/mL, 32 μg/ml and 64 μg/mL were 38%, 60%, 58%, 44%, 20%, and 100%, respectively. There are only 53 isolates in total, but there are too many subdivisions by the distribution of MICs in the study.

3. It is inappropriate to use cryptococcal antigen titer as indices of cryptococcus burden in the present study. Why not use CFU? And if CFU is not available in this study, please explain in the Limitations.

4. In this retrospective study, only 22.6% of patients were HIV-positive, but the overall in-hospital mortality rate is too high (50.9%), so the timing of medications, drug dosage, and the types of antifungal agents need to be provided. And the cause of death should be indicated? 

5. In this study, the authors believed that CM patients with fluconazole MIC ≥64 μg/mL showed extremely high mortality, which may suggest that the mortality rate was indeed high when fluconazole MIC value was ≥64 μg/mL, but only 2 CM patients with fluconazole MIC≥64 μg/mL, so it did not reflect the mortality rate that fluconazole MIC value can actually represent. According to the authors statement, can we also infer that the overall in-hospital mortality rate was very high when amphotericin B MIC was 0.125 μg/mL (100%, 1/1)? Therefore, the conclusion proposed by the authors is not reasonable.

Minor points

1. Please state the definition of wild type and non-wild type isolates?

2. Please state the criteria for SOFA score.

3. In line 228-229, the description about the practice of flucytosine monotherapy in the treatment of invasive cryptococcosis should be particularly discouraged due to the high resistance is unnecessary, because the guidelines clearly state that 5-FC cannot be treated alone because of high resistance.

Author Response

The manuscript attempted to investigate the correlation of antifungal susceptibility with prognosis of patients with cryptococcal meningitis . The authors reported concomitant cryptococcemia, low CSF glucose level, high CSF and serum cryptococcus antigen burden were risk factors for mortality. CM patients with fluconazole MIC ≥64 μg/mL showed extremely high mortality. However, the manuscript is not suitable to publication in current form and need to be substantially revised, because there are several limitations of this study, as shown below:

Major points

  1. The most major limitation of this study is that the sample size of isolates is too small to effectively reflect the relationship between antifungal susceptibility of isolates and prognosis.  

Reply: Dear reviewer, first we want to thank you for your kind review and comment. Your suggestion and advise help us improve our wok.The revised version of our article is now better than the original version. Here is my point to point reply to your comments.

we recognize this single center study had limited sample size (N=53). The small sample size might compromise the statistic power for results and may have led to an under-estimation of significant factors. We had described the limitation in article and remind our readers to take caution while analyzing the results. Nevertheless, we provided the MIC distribution of our cryptococcus isolates in the past years and tried our best to minimize the influence of confounding factors. We think this small study might be a pivot beginning for larger multicenter studies to investigate to build more robust results.

  1. The overall in-hospital mortality rates associated with amphotericin B MICs of 0.125 μg/mL, 0.25 μg/mL, 0.5 28 μg/mL, and 1 μg/mL were 100%, 50%, 51%, and 33%, respectively. The overall in-hospital mortality rates associated with fluconazole MICs of 2 μg/mL, 4 μg/mL, 8 μg/mL, 16 μg/mL, 32 μg/ml and 64 μg/mL were 38%, 60%, 58%, 44%, 20%, and 100%, respectively. There are only 53 isolates in total, but there are too many subdivisions by the distribution of MICs in the study.

Reply: We analyzed the mortality rates with individual distribution of MICs but found no trend of increasing mortalities by MIC stratification (p=0.37). The MICs of antifungal agent did not predict the outcomes of mortality. As for the concerns of too many MIC subdivisions, we also demonstrated the mortality rates based on only wild type (susceptible) and Non-wild type (resistance) in figure 1 and figure 2. And likewise, the difference did not reach conventional significance.

  1. It is inappropriate to use cryptococcal antigen titer as indices of cryptococcus burden in the present study. Why not use CFU? And if CFU is not available in this study, please explain in the Limitations.

Reply: Dear reviewer, we use cryptococcal antigen because the tests for detection of cryptococcal polysaccharide antigen in serum and CSF are extremely accurate for the diagnosis of invasive disease. The cryptococcal antigen titer, however, does give general prognostic information. Initial high titers (≥1:1024) demonstrate a high burden of yeasts in the host, poor host immunity, and a greater chance of therapeutic failure. (Reference : Mandell, Douglas, and Bennett's Principles and Practice of Infectious Diseases 2-Volume Set 9th Edition, Chapter 262 cryptococcosis). We do not apply CFU because it is encouraging when consideration of an IRIS diagnosis is being made that the cryptococcal antigen titer is stable or dropping. Finally, cryptococcal antigen titers can be used in follow-up concerning maintenance antifungal regimens but CFU cannot.

  1. In this retrospective study, only 22.6% of patients were HIV-positive, but the overall in-hospital mortality rate is too high (50.9%), so the timing of medications, drug dosage, and the types of antifungal agents need to be provided. And the cause of death should be indicated?

Reply: Dear reviewer, this is a small single center study. Some medical records were inevitably missing. Thus, we did not analyze the association between neurologic conditions and mortality outcomes. The timing of medications, drug dosage, and the therapeutic levels of antifungal agents were not included in the analysis. We have described this limitations in article, discussion section, line 273-276.

  1. In this study, the authors believed that CM patients with fluconazole MIC ≥64 μg/mL showed extremely high mortality, which may suggest that the mortality rate was indeed high when fluconazole MIC value was ≥64 μg/mL, but only 2 CM patients with fluconazole MIC≥64 μg/mL, so it did not reflect the mortality rate that fluconazole MIC value can actually represent. According to the authors’ statement, can we also infer that the overall in-hospital mortality rate was very high when amphotericin B MIC was 0.125 μg/mL (100%, 1/1)? Therefore, the conclusion proposed by the authors is not reasonable.

Reply: Dear reviewer, we very thank you for pointing out our unreasonable conclusion. In this study, we did not find a positive correlation between MICs and mortality. And the case number for amphotericin B 0.125ug/ml and fluconazole 64ug/ml were too small. We should not make the conclusion for high MIC relate to high mortality. We will delete this conclusion for new revision. Please refer to the revised version of article. However, the correlation between drug MIC and mortality remains uncertain. We try to remind clinical physicians when managing refractory or recurrent cryptococcal meningitis, a high fluconazole MIC should be considered for alternative antifungal agents.

Minor points

  1. Please state the definition of wild type and non-wild type isolates?

Reply: We have added definition in article section 2.4. Please refer to revised version.

The wild type population is that population of isolates/MICs in a species-drug combination with no detectable acquired resistance mechanisms. The highest wild type susceptibility endpoint has been defined as either the wild type cutoff value (COWT) or the Epidemiologic Cutoff Value, which is the critical drug concentration that may identify those strains with decreased susceptibility or serve as an early warning of emerging changes in the patterns of susceptibility of organisms to the agent being evaluated.

  1. Please state the criteria for SOFA score.

Reply: The Sequential Organ Failure Assessment (SOFA) score is a scoring system that assesses the performance of several organ systems in the body (neurologic, blood, liver, kidney, and blood pressure/hemodynamics) and assigns a score based on the data obtained in each category. The score is calculated based on Glasgow coma scale, mean arterial pressure or administration of vasopressors, PaO2/FiO2, platelet counts, bilirubin levels, and creatinine.

 The higher the SOFA score, the higher the likely mortality. The criteria for SOFA score are listed as below.

FORMULA

Addition of the selected points:

Variable

Points

PaO2/FiO2*, mmHg

≥400

0

300-399

+1

200-299

+2

≤199 and NOT mechanically ventilated

+2

100-199 and mechanically ventilated

+3

<100 and mechanically ventilated

+4

Platelets, ×103/µL

≥150

0

100-149

+1

50-99

+2

20-49

+3

<20

+4

Glasgow Coma Scale

15

0

13–14

+1

10–12

+2

6–9

+3

<6

+4

Bilirubin, mg/dL (μmol/L)

<1.2 (<20)

0

1.2–1.9 (20-32)

+1

2.0–5.9 (33-101)

+2

6.0–11.9 (102-204)

+3

≥12.0 (>204)

+4

Mean arterial pressure OR administration of vasoactive agents required (listed doses are in units of mcg/kg/min)

No hypotension

0

MAP <70 mmHg

+1

DOPamine ≤5 or DOBUTamine (any dose)

+2

DOPamine >5, EPINEPHrine ≤0.1, or norEPINEPHrine ≤0.1

+3

DOPamine >15, EPINEPHrine >0.1, or norEPINEPHrine >0.1

+4

Creatinine, mg/dL (μmol/L) (or urine output)

<1.2 (<110)

0

1.2–1.9 (110-170)

+1

2.0–3.4 (171-299)

+2

3.5–4.9 (300-440) or UOP <500 mL/day)

+3

≥5.0 (>440) or UOP <200 mL/day

+4

*Estimating FiOâ‚‚ from oxygen flow/delivery rates:

Type of Oâ‚‚ delivery

Flow rates, L/min

FiOâ‚‚

Nasal cannula

1-6

~4% FiOâ‚‚ added above room air* per 1 L/min

  • Room air = 21% 
  • 1 L/min = 25%
  • 2 L/min = 29%
  • 3 L/min = 33%
  • 4 L/min = 37%
  • 5 L/min = 41%
  • 6 L/min = 45%

Simple face mask

~6-12

35-60%*

Non-rebreather mask

10-15

~70-90%

High-flow nasal cannula

Up to 60

30-100%

  1. In line 228-229, the description about “the practice of flucytosine monotherapy in the treatment of invasive cryptococcosis should be particularly discouraged due to the high resistance” is unnecessary, because the guidelines clearly state that 5-FC cannot be treated alone because of high resistance.

Reply: The description is deleted in revised version. As a result, our data is correspondent with guideline suggestions.

Round 2

Reviewer 2 Report

Consider changing Cryptococcus to Cryptococcus throughout the text.

Reviewer 3 Report

I am not satisfied with the authors' reply to my comments, especially for major points 3 and 4. The authors are not willing to modify or supplement the corresponding data.

In fact, during the antifungal treatment, CrAg titer can be kept at the same level as the baseline for a period of time, but its fungal burden (CFU, as indices of viable cryptococcus) declines after treatment.

Because of the high mortality rate of this study, the authors should provide detailed data on treatment and causes of death. The author replied that these data are missing, however the mortality provided by the authors is the overall mortality in hospital, and these data should not be missing.